# Treating Primary Node-Positive Prostate Cancer: A Scoping Review of Available Treatment Options

**DOI:** 10.3390/cancers15112962

**Published:** 2023-05-29

**Authors:** Lotte G. Zuur, Hilda A. de Barros, Koen J. C. van der Mijn, André N. Vis, Andries M. Bergman, Floris J. Pos, Jeroen A. van Moorselaar, Henk G. van der Poel, Wouter V. Vogel, Pim J. van Leeuwen

**Affiliations:** 1Department of Surgical Oncology (Urology), Netherlands Cancer Institute, 1066 CX Amsterdam, The Netherlands; l.g.zuur@nki.nl (L.G.Z.); h.d.barros@nki.nl (H.A.d.B.); h.vd.poel@nki.nl (H.G.v.d.P.); 2Department of Medical Oncology, Netherlands Cancer Institute, 1066 CX Amsterdam, The Netherlands; k.vd.mijn@nki.nl (K.J.C.v.d.M.); a.bergman@nki.nl (A.M.B.); 3Department of Urology, Amsterdam University Medical Centre, 1081 HV Amsterdam, The Netherlands; a.vis@amsterdamumc.nl (A.N.V.); rja.vanmoorselaar@amsterdamumc.nl (J.A.v.M.); 4Department of Radiation Oncology, Netherlands Cancer Institute, 1066 CX Amsterdam, The Netherlands; f.pos@nki.nl (F.J.P.); w.vogel@nki.nl (W.V.V.); 5Department of Nuclear Medicine, Netherlands Cancer Institute, 1066 CX Amsterdam, The Netherlands

**Keywords:** prostate cancer, primary diagnosed, node-positive, metastasis, treatment, radiotherapy, androgen deprivation therapy, radical prostatectomy

## Abstract

**Simple Summary:**

The best way to treat patients with prostate cancer who have positive lymph nodes is not clear. However, recent studies have suggested that intensifying treatment may help these patients and potentially cure their cancer. This review summarises the existing research that supports various treatment options investigated for these patients. For patients with positive lymph nodes that can be seen on pre-treatment imaging (clinically node-positive), the best treatment option is a combination of hormonal therapy and radiotherapy. Although intensifying the treatment seems promising, more research studies are needed to confirm its effectiveness. For patients with positive lymph nodes confirmed through pathology tests (pathologically node-positive), the treatment options depend on evaluating the risks based on factors such as the Gleason score, tumour stage, the presence of positive lymph nodes, and surgical margins. These patients should be closely monitored, and it is recommended to consider additional treatment with hormonal agents and/or radiotherapy.

**Abstract:**

There is currently no consensus on the optimal treatment for patients with a primary diagnosis of clinically and pathologically node-positive (cN1M0 and pN1M0) hormone-sensitive prostate cancer (PCa). The treatment paradigm has shifted as research has shown that these patients could benefit from intensified treatment and are potentially curable. This scoping review provides an overview of available treatments for men with primary-diagnosed cN1M0 and pN1M0 PCa. A search was conducted on Medline for studies published between 2002 and 2022 that reported on treatment and outcomes among patients with cN1M0 and pN1M0 PCa. In total, twenty-seven eligible articles were included in this analysis: six randomised controlled trials, one systematic review, and twenty retrospective/observational studies. For cN1M0 PCa patients, the best-established treatment option is a combination of androgen deprivation therapy (ADT) and external beam radiotherapy (EBRT) applied to both the prostate and lymph nodes. Based on most recent studies, treatment intensification can be beneficial, but more randomised studies are needed. For pN1M0 PCa patients, adjuvant or early salvage treatments based on risk stratification determined by factors such as Gleason score, tumour stage, number of positive lymph nodes, and surgical margins appear to be the best-established treatment options. These treatments include close monitoring and adjuvant treatment with ADT and/or EBRT.

## 1. Introduction

Prostate cancer (PCa) is the second most commonly diagnosed cancer in men worldwide [1]. The treatment of patients with PCa depends on the stage and grade of the disease and often involves a multidisciplinary approach. The presence of nodal metastasis (N+) is an unfavourable prognostic factor that correlates with PCa recurrence, distant metastases, and survival [2]. Accounting for approximately 13% of newly diagnosed PCa cases, N+ PCa contributes significantly to the total number of deaths from PCa [1].

Historically, patients diagnosed with primary lymph-node-positive PCa received androgen deprivation therapy (ADT) alone as their primary treatment. Patients with nodal PCa metastases were not considered for further curative treatment due to the assumption that a cure was not feasible. This was based on the belief that patients with lymph-node-positive disease suffer from systemic disease. However, these assumptions are being questioned today. Several studies have shown that a radical prostatectomy (RP) combined with extended pelvic lymph node dissection (LND) without adjuvant therapy can cure patients with one or two positive lymph nodes (pN1M0) (Cancer-Specific Survival (CSS) of >95% after 5 years) [3,4]. The same finding has been demonstrated for external beam radiotherapy (EBRT) applied to the prostate and regional lymph nodes in combination with concurrent hormone therapy [5,6]. Moreover, new molecular imaging techniques targeting prostate-specific membrane antigen (PSMA) detect smaller nodal metastases while reliably excluding distant metastases with a high positive predictive value [7]. As a result, more patients are diagnosed with node-positive PCa without distant metastases (cN1M0). These findings have increased confidence that N+ PCa patients may be curable and eligible for loco-regional therapy, which could be combined with systemic treatment [8]. These intensifications of treatment may be beneficial, but they can also increase the occurrence of adverse events and potentially impair quality of life. Due to the limited availability of randomised data, current guidelines are hesitant to recommend the implementation of these intensified treatments [2,9,10] (Table 1). Consequently, the management of this patient population varies widely in clinical practice, and the optimal treatment remains a topic of discussion.

This scoping review aims to provide an overview of the current evidence on the available treatments for men with primary diagnosed clinically and/or pathologically node-positive PCa. Our goal is to guide clinical care and further scientific research.

## 2. Materials and Methods

This scoping review was performed in accordance with the Preferred Reporting Items for Systematic Reviews and Meta-Analyses (PRISMA) guideline on scoping reviews [11]. Medline was searched for relevant clinical studies published in the English language from January 2002 to December 2022 using the following terms: “Primary Prostate Cancer AND Nodal Metastasis”, “Primary Prostate Cancer AND Nodal Metastasis AND randomised controlled trials (RCT)”, and “Primary Prostate Cancer AND Nodal Metastasis AND Systematic Reviews/Meta-Analyses”. Clinicaltrials.gov was searched for ongoing clinical trials (Figure 1).

Primary-diagnosed node-positive PCa patients fall into two categories: patients with clinically node-positive disease (cN1M0) and patients with pathologically node-positive disease (pN1M0). For this scoping review, we focused on cN1M0 patients that were staged as node-positive on any imaging modality (i.e., computed tomography (CT)/magnetic resonance imaging (MRI)/bone scan/prostate-specific membrane antigen positron emission tomography (PSMA-PET)/CT)) without prior treatment and on pN1M0 patients that were staged as node-positive in a postoperative setting either after RP and LND or after staging LND.

The Population, Intervention, Comparator, Outcome, and Study (PICOS) design model was used as a guide for eligibility and was constructed twice due to the two patient categories. Both constructed models are shown in Table 2 and Table 3.

Reports were considered relevant to this scoping review if they involved patients with either cN1M0 disease (Table 2) or pN1M0 disease (Table 3). Furthermore, reports were considered if they compared two different types of treatment to determine oncologic outcomes in terms of survival for either of the patient categories.

We selected studies according to the following criteria: (1) treatment and outcomes of cN1M0/pN1M0 PCa patients; (2) provision of original data, including RCTs and retrospective studies (no editorial notes); (3) reporting of consecutive cases (no case reports); and (4) provision of systematic reviews and meta-analyses.

The screening consisted of scanning titles and abstracts with regard to their relevance for inclusion. Afterwards, the remaining full-text original articles were retrieved. Abstracts and original articles were reviewed by one reviewer for eligibility. 

## 3. Results

### 3.1. Treatment of Patients with cN1M0 Disease

Table 4 lists the included studies evaluating the different treatments for cN1M0 disease, which have been divided into subcategories: (1) treatment with ADT alone, (2) treatment with only local therapy, (3) treatment with ADT combined with any form of local therapy, and (4) treatment with ADT and other forms of systemic treatment.

#### 3.1.1. Treatment with ADT

ADT has long been considered as the standard of care and is still recommended in guidelines for cN1M0 disease [2,9,10] (Table 1). The timing of starting ADT has been investigated by Schröder et al. in the EORTC 30846 study [12]. Patients with lymph-node-positive PCa confirmed after staging LND without local treatment were randomised to immediate ADT or ADT provided at the time of clinical progression, in which the delayed group received ADT for 2.7 years and the immediate group received ADT for 3.2 years. After a 13-year follow-up, the median overall survival (OS) was 7.6 years for the immediate ADT arm and 6.1 years for the delayed ADT arm. The intention-to-treat analysis did not show a statistically significant difference in survival between an immediate or delayed start of ADT (hazard ratio (HR) = 1.22; 95% confidence interval (CI): 0.92–1.62).

#### 3.1.2. Treatment with Local Therapy

Two studies have been conducted on treatment with local therapy (LT) (i.e., EBRT applied to both prostate and lymph nodes or RP with LND), with or without the addition of ADT. The phase III study of RTOG 85-31 randomised patients undergoing treatment with EBRT either with the addition of ADT or without [13]. After ten years, the study group with combination treatment demonstrated an extended absolute survival compared to the local therapy group (*p* = 0.002). Tward et al. used the SEER database to retrospectively analyse patients treated without EBRT, with EBRT, or with EBRT plus brachytherapy [14]. Patients treated with EBRT had an increased 10-year cancer-specific survival (CSS) rate compared to patients not treated with EBRT (HR 0.66, 95% CI 0.54–0.82, *p* < 0.01).

#### 3.1.3. Treatment with ADT ± Any Form of Local Therapy

Since the RTOG 85-31 study showed positive results in terms of absolute survival in favour of the combined treatment ADT + EBRT (49% ADT + EBRT vs. 39% EBRT alone, *p* = 0.002) [13], multiple trials have investigated the role of ADT in combination with definitive EBRT in node-positive PCa. Another four studies on definitive EBRT for node-positive PCa consistently showed improved OS and cancer-specific survival (CSS) compared to ADT only or conservative management only (Table 4) [15,16,17,19]. Most of these articles reported that the radiation fields included both prostate and lymph nodes; however, studies employing the SEER database did not provide specific details regarding radiation fields. 

ADT in combination with any form of local treatment (i.e., RP or EBRT) has also retrospectively been investigated by Seisen et al., who reported a survival advantage for treatment with ADT in combination with any form of local therapy [18]. Patients who underwent RP were statistically younger (with a mean age of 61.3) compared to patients undergoing EBRT (with a mean age of 65.8) (*p* ≤ 0.001). However, this study showed no statistically significant differences between RP and EBRT. 

#### 3.1.4. Treatment with ADT ± Additional Systemic Therapy

It is known that combining ADT with docetaxel or second-generation hormone treatment improves the outcome of metastatic PCa [21,22,23,24]. However, until recently, none of these drugs have demonstrated a clear and consistent improvement in the survival of patients with non-metastatic PCa starting palliative ADT [25]. In three trials, one of which was conducted in the STAMPEDE platform protocol, another in the NRG Oncology/RTOG 0512 trial, and the third in the GETUG-12 trial, adjuvant docetaxel added to ADT prolonged time to relapse but not metastasis-free survival or OS. A meta-analysis of these adjuvant docetaxel trials incorporating N0/N1-M0 patients concluded that there was an 8% absolute 4-year survival advantage for docetaxel compared with ADT alone in terms of failure-free survival without an OS benefit (HR 0.7, 95% CI 0.61–0.81; *p* < 0.0001) [20]. Collectively, these results indicate that docetaxel does not offer a benefit in terms of OS for patients with cN1M0 disease.

More recently, a meta-analysis of two STAMPEDE platform phase III trials found that the addition of abiraterone acetate and prednisolone with or without enzalutamide to ADT was associated with improved metastasis-free survival in patients with high-risk nonmetastatic prostate cancer [8]. Thirty-nine percent of the patients (n = 774) presented a cN1 status determined via conventional imaging. Of these patients, around 85% received EBRT and ADT as a standard-of-care treatment. Metastasis-free survival events occurred for 180 patients in the combination groups vs. 306 in the control groups (HR 0.53; 95%CI 0.44–0.64, *p* < 0.0001). Death occurred in 147 patients in the combination groups vs. 236 in the control groups (HR 0.60; 95% CI = 0.48–0.73, *p* < 0.0001). Death due to PCa occurred in 73 patients in the combination groups vs. 142 in the control groups (HR 0.49; 95% CI = 0.37–0.65, *p* < 0.0001). These results indicate that the addition of abiraterone and/or enzalutamide may be a promising treatment option for cN1M0 patients, offering potential benefits for overall survival. However, since this is a post-hoc analysis and meta-analysis, it is crucial to approach these findings with caution when drawing conclusions.

### 3.2. Treatment of Patients with pN1M0 Disease

Table 5 lists the included studies evaluating the different treatments for pN1M0 disease, which have been divided into subcategories: (1) ADT as an adjuvant treatment, (2) ADT with or without EBRT as an adjuvant treatment, (3) EBRT as an adjuvant treatment, and (4) chemotherapy as adjuvant treatment.

#### 3.2.1. ADT as Adjuvant Treatment

The ECOG 3886 trial is the only randomised trial that investigated the use of an adjuvant treatment with ADT. This trial randomised 98 patients who were proven to have a pN1 status after RP and LND for immediate ADT or delayed ADT [26]. After a median follow-up of 11.9 years, the trial showed that immediate ADT resulted in statistically better OS (HR 1.84, 95%CI 1.01–3.35, *p* = 0.04) and CSS (HR 4.09, 95%CI 1.76–949, *p* = 0.0004). Despite its small sample size, the ECOG 3886 trial provides the only level 1 evidence for an OS benefit of adjuvant treatment for pathologically positive lymph node patients after radical prostatectomy.

Subsequently, the observation of patients with pN1M0 disease was retrospectively analysed in three studies. These studies compared observation to adjuvant treatment with ADT or ADT combined with EBRT [5,27,28]. Touijer et al. found that adjuvant ADT treatment resulted in better CSS when compared to observation (HR: 0.64, 95%CI 0.43–0.95, *p* = 0.027) but that OS was similar (HR 0.90, 95% CI 0.65–1.25, *p* = 0.5) due to the association of ADT with an increased risk of other-cause mortality compared with observation (HR 3.05, 95%CI 1.45–6.40, *p* = 0.003). However, they also reported 28% BCR-free survival at the 10-year mark among men with pN1M0 without any adjuvant therapy [5]. Wong et al. and Gupta et al. reported similar results in favour of adjuvant ADT [27,28].

#### 3.2.2. ADT ± EBRT as Adjuvant Treatment

The role of adjuvant EBRT in combination with ADT for pN1M0 disease has been retrospectively investigated. Seven studies found improved survival compared to ADT alone, with an HR for OS ranging from 0.46 to 0.77, while other studies used different end points, such as 10-year CSS ranging from 70% to 86% [5,6,27,28,29,31,32]. Contrarily, one study showed no benefit (*p* = 0.193) [30]. These poor-quality data emphasise the heterogeneity of this patient population, presenting different outcomes depending on the tumour grade or the number of positive lymph nodes.

Similarly, Van Hemelryk et al. performed a matched-case analysis of pN1M0 and pN0M0 to compare outcomes after EBRT + ADT and found promising survival rates, especially for patients with two or fewer positive lymph nodes (5 yr biochemical relapse-free survival (bRFS) of 65% (pN1) vs. 79% (pN0) (*p* = 0.08)) [33]. In addition, Abdollah et al. also found that this specific risk group of patients benefitted the most from adjuvant EBRT along with patients with a Gleason score ranging from seven to ten, a pathological tumour stage (pT) of 3b/4, positive surgical margins, or with a pelvic lymph node count of three to four [31]. 

One observational study compared the different types of local therapy (i.e., RP or EBRT) in the case of pN1M0 disease after staging LND [15]. Rusthoven et al. observed no statistically significant differences in survival between RP versus EBRT and RP with or without adjuvant EBRT.

#### 3.2.3. EBRT as Adjuvant Treatment

The PROPER trial prospectively compared whole-pelvis radiotherapy (WPRT) to prostate-only radiotherapy (PORT) in the case of pN1M0 disease and found non-statistically significant differences in clinical relapse-free survival (cRFS), bRFS, and OS in favour of WPRT (*p* = 0.31, *p* = 0.08, and *p* = 0.61, respectively) [35]. The study included a total of 64 patients and was closed early due to poor accrual. 

Tilki et al. compared adjuvant EBRT with early salvage EBRT, both applied to the prostatic bed, and found statistically significantly lower all-cause mortality in pN1M0 patients treated with adjuvant EBRT (HR 0.66, 95%CI 0.44–0.99; *p* = 0.04) [36].

#### 3.2.4. Chemotherapy as Adjuvant Treatment

The SPCG-12 trial randomised patients after RP into groups of either treatment with docetaxel or a surveillance group [37]. There was no improvement in biochemical disease-free survival (*p* = 0.06). Details on metastasis-free survival (MFS) or OS were not further described.

## 4. Discussion

The optimal treatment for clinically or pathologically node-positive PCa patients remains poorly defined. As there is mounting evidence that these patients may benefit from treatment intensification, additional treatment modalities are being increasingly implemented in conjunction with the standard ADT treatment alone. However, because there is limited evidence from randomised controlled trials, many of the recommendations in international guidelines are considered weak. Hence, we conducted a scoping review to summarise the current evidence on available treatments. 

First, we have analysed the available studies on men with cN1M0 PCa. The combination of ADT and EBRT to treat both the prostate and lymph nodes in these patients has been well established. Several studies have shown that combined treatment with ADT and EBRT provides a greater survival benefit compared to treatment with ADT or EBRT alone. The optimal duration of ADT has not been well-defined, with data supporting 18 to 36 months, while in practice, 2 to 3 years are frequently recommended by the United States and European guidelines, respectively [2,9,38]. The treatment outcomes for men with cN1M0 PCa are comparable to those of patients with de novo metastatic hormone-sensitive PCa with low-burden disease. This assumption has been made because it is believed that most men with cN1M0 PCa have microscopic distant metastases. Based on the results of two randomised controlled trials [39,40] and one meta-analysis [41], international guidelines strongly recommend the combination of ADT plus EBRT applied to the prostate in de novo metastatic hormone-sensitive PCa with low-burden disease, according to the CHAARTED criteria [42]. Nevertheless, none of the EBRT studies included in this scoping review differentiated between WPRT or PORT. This differentiation has been studied in men with high-risk cN0 disease in the RTOG 9413, GETUG-01, and POP-RT trials. The RTOG 9413 and GETUG-01 trials did not show a statistically significant advantage of WPRT over PORT [43,44]. However, the POP-RT trial reported better outcomes in terms of biochemical failure-free survival (BFFS) and disease-free survival (DFS) for men who underwent WPRT [45]. A systematic review conducted by De Merleer et al. on elective EBRT suggested a simultaneous boost to PSMA-PET-positive nodes; nevertheless, there are scarce randomised data supporting this suggestion [46]. 

One of the most recent comparisons in the STAMPEDE trial evaluated the addition of abiraterone with or without enzalutamide to ADT in men with locally advanced hormone-sensitive PCa in two separate studies [8]. Data from both trials were pooled together and published. Newly diagnosed patients were randomly assigned to receive 36 months of ADT with or without an additional 24 months of abiraterone treatment. The addition of abiraterone (plus/minus enzalutamide) to ADT and EBRT significantly improved MFS and OS. However, these results should be considered with caution due to their post-hoc and meta-analytical nature. The STAMPEDE trial incorporating high-risk non-metastatic hormone-sensitive PCa patients may change clinical practice by providing evidence for the addition of two years of abiraterone therapy to ADT and EBRT for men with newly diagnosed cN1M0 PCa. Based on the clinically significant benefits in terms of MFS and OS seen in the STAMPEDE trial, the EAU guideline panel recommends the addition of two years of abiraterone therapy to ADT as the standard of care for men receiving EBRT for high-risk disease as defined by the STAMPEDE criteria (cN+ or two of the following: a Gleason grade ≥8, a PSA level ≥40 ng/Ml, and/or ≥cT3) (Table 1). Nevertheless, it is crucial to account for the significant factors when considering abiraterone in addition to ADT and EBRT for men with cN1M0 PCa. Imaging in the STAMPEDE trial consisted of MRI, CT, and bone scans, whereas PSMA-PET/CT is now being increasingly used for staging purposes. Whether abiraterone or other androgen receptor agents may benefit a patient with <5 mm pelvic nodal disease on PSMA PET/CT imaging (i.e., not measurable on a conventional scan) is uncertain and requires further investigation in future studies. Since the combination of abiraterone and enzalutamide did not improve outcomes compared to abiraterone alone and the toxicity was greater with the combination, it is not recommended in these patients. Enzalutamide has shown to be beneficial in the treatment of metastatic castration-resistant PCa (mCRPC), especially in low-volume disease [21]. However, its usefulness in cN1M0 disease has not yet been thoroughly investigated. Nonetheless, there may be potential for enzalutamide to be useful in treating cN1M0 disease considering its positive results with respect to low-volume mCRPC. Further research is needed to fully understand the potential benefits of enzalutamide in cN1M0 disease. Results are pending from ongoing trials evaluating other novel hormonal agents (NCT 04134260) and the use of other treatment modalities such as lutetium (NCT 05162573). With the emergence of advanced diagnostic techniques such as PSMA PET-CT and the findings from recent studies, it is reasonable to anticipate a shift and update in the treatment of patients with cN1M0 PCa.

Second, we have studied the available treatments for men with pN1M0 PCa. A risk-adapted strategy for selecting pN1 PCa patients for adjuvant or early salvage therapy may be the most effective approach. In a randomised controlled trial, Messing et al. showed that immediate ADT in pN1 patients improves the survival [26]. However, the study included long-term survivors among the patients in the control arm, which raises the question of whether adjuvant treatment, inducing significant side effects, is necessary for all patients. Several analyses have shown that the number of lymph node metastases is correlated with survival outcomes. Patients with limited lymph node metastases tend to have better outcomes [4,5,47]. Moreover, several studies have shown that RP combined with extended pelvic lymph node dissection without adjuvant therapy can be curative for some patients with lymph-node-positive PCa [4,5,47]. De Meerleer et al. recommended adjuvant WPRT plus ADT for pN1M0 patients with two to four positive lymph nodes [46]. This recommendation is mainly based on the findings presented by Touijer et al., who conducted a study comparing three different adjuvant treatment strategies (observation, ADT, or EBRT + ADT) among men with pN1M0. The study revealed that men who received EBRT + ADT had significantly better OS compared to those who were treated with observation or ADT after RP only [5]. 

Collectively, these findings underscore the importance of selecting patients based on their pathological characteristics and conducting regular follow-ups with PSA testing to determine whether adjuvant treatment is required. Patients with a Gleason score ≥8, positive surgical margins, pT ≥ 3b, or ≥3 positive lymph nodes benefit the most from adjuvant treatment. Conversely, patients who have a lower risk of recurrence may benefit from active surveillance and receiving treatment only if their PSA level rises. Evidence on the timing of post-prostatectomy EBRT has not been studied for pN1M0 disease, but randomised data in the case of localised disease have failed to show a survival benefit for adjuvant EBRT compared to salvage EBRT [48,49]. This might suggest that patient selection is also key in determining the timing of adjuvant or early salvage therapy. 

The majority of the studies included in this review used conventional imaging techniques, such as CT, MRI, and bone scans, for staging. However, these methods have been shown to have limited sensitivity in detecting lymph node metastases, which can potentially lead to an underestimation of the disease’s extent, especially for cN1M0 disease. Recent evidence suggests that PSMA-PET/CT is more accurate than conventional imaging for (re)staging PCa [7,50,51]. Therefore, the results of studies in which patients were staged using conventional imaging cannot be compared with recent studies in which staging was performed with the use of PSMA-directed tracers. Moreover, enhanced accuracy in staging, which can be facilitated by the utilisation of PSMA-targeted tracers, will result in improved patient selection. Hence, the outcomes of research utilising PSMA-targeted tracers to determine a patient’s disease stage are eagerly anticipated as they have the potential to significantly influence treatment recommendations.

There are several significant limitations to this scoping review. First, it should be noted that this review is not a systematic review and, therefore, was not registered online. Articles were not double-selected, and data were not pooled or used for a meta-analysis. However, the PRISMA guidelines for scoping reviews were followed to ensure a systematic approach to conducting this scoping review. 

Second, most studies were conducted in a retrospective setting and thus were subject to all the inherent limitations that come with this approach. Additionally, national databases (Surveillance, Epidemiology, and End Results (SEER) and the National Cancer Database (NCDB)) were used by multiple retrospective studies, which not only resulted in overlapping cohorts of patients but also led to low granularity and heterogeneity in exposure ascertainment. 

Third, this scoping review does not delve into the discussion of treatment toxicities. The decision was made to focus primarily on oncological outcomes due to the extensive scope of the papers being reviewed. Nonetheless, considering the potential usefulness of treatment, exploring the toxicity profiles, particularly in combination therapies, can provide valuable insights. Furthermore, an in-depth exploration of patients’ comorbidities in relation to treatments could have shed light on whether certain treatments may act as protective factors or risk factors for the progression of oncological disease.

Trials including novel staging modalities, biomarkers, new antiandrogen drugs, or other treatment modalities are currently underway and are necessary for providing high-quality evidence to guide treatment decisions. It is probable that these studies will provide additional insights into the treatment of patients with clinically or pathologically node-positive PCa.

## 5. Conclusions

This study presents a scoping summary of the evidence on the treatment of clinically and pathologically node-positive PCa patients. Combined treatment with EBRT applied to the prostate and lymph nodes, along with ADT, is a well-established and effective treatment for cN1M0 disease. There is evidence that treatment intensification can be beneficial, but further randomised studies are needed to confirm this more conclusively. 

In the case of pN1M0 disease, the corresponding oncological control and survival rates are encouraging, as a significant percentage of patients remain disease-free. Based on retrospective studies, adjuvant EBRT combined with ADT has been shown to improve so-treated patients’ overall survival compared to men who were treated with observation or ADT alone. Patient selection is crucial in this case, in which patients with a Gleason score of ≥8, positive surgical margins, pT ≥ 3b, or ≥3 positive lymph nodes benefit the most from adjuvant treatment.

## Figures and Tables

**Figure 1 cancers-15-02962-f001:**
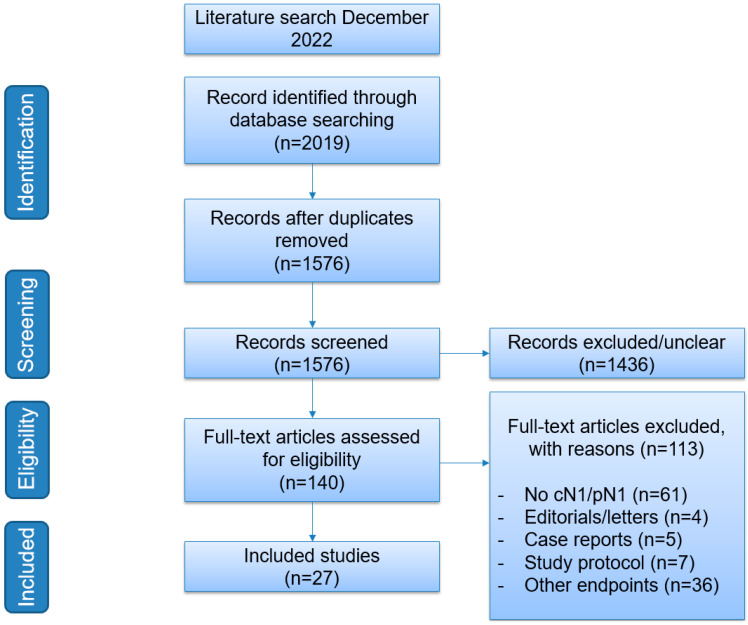
Flowchart of search strategy.

**Table 1 cancers-15-02962-t001:** Current guidelines on management of N1M0 PCa.

Guideline	cN1M0	pN1M0
EAU [2]	1 Offer local treatment (either RP or EBRT) plus long-term ADT 2 Offer EBRT for prostate + pelvis in combination with long-term ADT and 2 years of abiraterone	1 Offer adjuvant ADT 2 Offer ADT + EBRT 3 Offer observation (expectant management) after LND if ≤ 2 nodes and PSA < 0.1 ng/mL
FROGG [10]	1 Pelvis and prostate EBRT + long-term ADT	1 Individualised discussion of observation, ADT, or EBRT + ADT 2 Patients should be referred to a radiation oncologist to discuss EBRT + ADT
NCCN [9]	1 EBRT + ADT 2 EBRT + ADT + abiraterone 3 ADT ± abiraterone 4 If <5 yr expected survival and asymptomatic: observation or ADT	1 ADT 2 EBRT + ADT 3 Observation

Abbreviations: cN1M0 patients = patients with clinically node-positive disease determined via any imaging modality; pN1M0 patients = patients with pathologically node-positive disease in postoperative setting either after initial treatment with radical prostatectomy (RP) and lymph node dissection (LND) or after staging LND; PCa = prostate cancer; EBRT = external beam radiotherapy; ADT = androgen deprivation therapy; PSA = prostate specific antigen.

**Table 2 cancers-15-02962-t002:** The applied Population, Intervention, Comparator, Outcome, and Study model (PICOS) for cN1M0.

P	Population	Patients with clinically node-positive prostate cancer (cN1M0)
I	Intervention	Treatment A
C	Comparator	Treatment B
O	Outcome	Oncological outcome, survival

**Table 3 cancers-15-02962-t003:** The applied Population, Intervention, Comparator, Outcome, and Study model (PICOS) for pN1M0.

P	Population	Patients with pathologically node-positive prostate cancer (pN1M0) after primary lymph node dissection (RP + LND as treatment or with staging LND)
I	Intervention	Treatment A
C	Comparator	Treatment B
O	Outcome	Oncological outcome, survival

**Table 4 cancers-15-02962-t004:** Overview of studies on cN1M0 disease.

	Study Design	Number of cN1M0 Patients	Treatment Groups	Study Period	Outcome
ADT					
Schröder et al. (2009) [12]	RCT: EORTC 30846	234	Arm I: immediate ADT ARM II: delayed ADT	1986–1998	No statistically significant differences Median OS 7.6 yr (95% CI, 6.3–8.3 yr) (immediate) vs. 6.1 yr (95% CI, 5.7–7.3 yr) (delayed) HR 1.22 for OS (95% CI 0.92–1.62); not statistically significant
Local therapy					
Pilepich et al. (2005) [13]	RCT: Phase III RTOG 85-31	263	Arm I: EBRT + goserelin Arm II: RT alone	1987–1992	Favours EBRT + ADT, especially high GS. 10 yr absolute survival 49% (RT + ADT) vs. 39% (RT alone) (*p* = 0.002)
Tward et al. (2013) [14]	Observational, SEER database	1100	EBRT vs. no EBRT	1988–2006	Favours EBRT 10-yr CSS 50% vs. 63%; (HR 0.66, 95% CI 0.54–0.82, *p* < 0.01) 10-yr OS 29% vs. 44%; (HR 0.70, 95% CI 0.59–0.81, *p* < 0.01)
ADT ± any form of local therapy					
Rusthoven et al. (2014) [15]	Observational, SEER database	796	ADT vs. ADT + EBRT	1995–2006	Favours EBRT over no therapy 10 yr OS rate of 45% vs. 29% (*p* < 0.001) 10 yr PCSS rate 76% vs. 53% (*p* < 0.001)
Lin et al. (2015) [16]	Observational, NCDB	3540 total—318 propensity scored matched	ADT vs. ADT + EBRT	2004–2011 Sub-cohort 2004–2006	Favours ADT + EBRT 50% reduction in 5-yr all-cause mortality (HR = 0.50, 95% CI 0.37–0.67, *p* < 0.001)
James et al. (2016) [17]	Observational, control arm of STAMPEDE	177	ADT vs. ADT + EBRT	2005–2014	Favours ADT + EBRT HR 0.48 (95% CI 0.29–0.79) 2 yr FFS of 81% (95% CI 0.71–0.85) (ADT + RT) vs. 53% (95% CI 0.40–0.65) (ADT)
Seisen et al. (2018) [18]	Observational, NCDB	1987	ADT vs. ADT + Local Therapy (LT)	2003–2011	Favours ADT + LT 5 yr OM-free survival was 78.8% (95% CI 74.1–83.9%) (ADT + LT) vs. 49.2% (95% CI 33.9–71.4) (ADT)
Bryant et al. (2018) [19]	Observational, Veterans Affair database	648	ADT vs. ADT + EBRT	2000–2015	Favours ADT + EBRT PSA < 26: ADT + RT improved PCSM (HR 0.50: 95%CI 0.28–0.88; *p* = 0.02) and ACM (HR 0.38; 95%CI 0.25–0.57; *p* < 0.001)
ADT ± Systemic treatment					
Vale et al. (2016) [20]	Systematic review: GETUG-12, RTOG 0521, STAMPEDE	945	ADT ± docetaxel	2002–2013	OS: No benefit adding docetaxel: HR 0.87, 95%CI 0.69–1.09; *p* = 0.218)
Attard et al. (2022) [8]	RCT 1: abiraterone trial RCT 2: abiraterone + enzalutamide trial STAMPEDE protocol	774	1: ADT (control) vs. ADT + abiraterone and prednisolone (combi-therapy group) 2: ADT + (control) vs. ADT + abiraterone + prednisolone + enzalutamide (combi-therapy group)	2011–2016	Favours ADT + abiraterone + prednisolone 6 yr metastasis-free survival 82% (95%CI 79–85) (combination therapy) vs. 69% (95% CI 66–72) (control); HR 0.53, 95% CI 0.44–0.64, *p* < 0.0001)

Abbreviations: cN1M0 patients = patients with clinically node-positive disease on any imaging modality; ADT = androgen deprivation therapy; EBRT = external beam radiotherapy; RP = radical prostatectomy; LT= local therapy (i.e., EBRT or RP); RCT = randomised controlled trial; SEER = Surveillance, Epidemiology, and End Results; NCDB = National Cancer Database; OS = overall survival; CI = confidence interval; HR = hazard ratio; GS = Gleason score; yr = year; CSS = cancer-specific survival; PCSS = prostate-cancer-specific survival; FFS = failure-free survival; OM-free = overall mortality free; PSA = prostate-specific antigen; PCSM = prostate-cancer-specific mortality; ACM = all-cause mortality;; DE = docetaxel and estramustine.

**Table 5 cancers-15-02962-t005:** Overview of studies on pN+ disease.

	Study Design	Number of pN1M0 Patients	Treatment Groups	Study Period	Outcome
ADT as adjuvant treatment					
Messing et al. (2006) [26]	RCT ECOG 3886	98	Immediate ADT vs. deferred ADT	1988–1993	Favour immediate ADT:Superior OS for immediate adjuvant ADT (HR 1.84, 95% CI 1.01–3.35, *p* = 0.04). Superior PCSS for immediate ADT (HR 4.09 (95%CI 1.76–9.49), *p* = 0.0004)
Wong et al. (2016) [27]	Retrospective, NCDB	7225	No adjuvant therapy ADT alone EBRT alone ADT + EBRT	2004–2011	Favours adjuvant EBRT + ADT: 5-yr OS rate, 85.2% (no therapy), 82.9% (ADT), 88.3% (EBRT), 88.8% (ADT + EBRT) (*p* ≤ 0.001)
Touijer et al. (2018) [5]	Retrospective, three institutions	1338	Observation vs. ADT alone vs. ADT + EBRT	1988–2010	CSS: Favours ADT compared to observation (HR: 0.64, 95% CI: 0.43–0.95, *p* = 0.027). OS: similar between ADT and observation, due to ADT increased risk of other-cause mortality (HR: 3.05, 95% CI: 1.45–6.40, *p* = 0.003)
Gupta et al. (2019) [28]	Retrospective	8074	Observation vs. ADT vs. ADT + EBRT	2004–2013	No difference in OS between ADT vs. observation (HR 1.01, 95%CI 0.87–1.18, *p* = 0.88)
ADT ± RT as adjuvant treatment					
Da Pozzo et al. (2009) [6]	Retrospective, single institution	250	ADT vs. ADT + EBRT	1988–2002	Favours adjuvant EBRT: 10 yr BCR-free survival 51% (ADT + EBRT) vs. 42% (ADT) (*p* = 0.11) 10 yr CSS rate 70% (ADT + EBRT) vs. 72% (ADT) (*p* = 0.22)
Briganti et al. (2011) [29]	Retrospective, two institutions	364	ADT vs. ADT + EBRT (matched analysis)	1986–2002	Favours adjuvant EBRT: 10 yr CSS 86% (ADT + EBRT) vs. 70% (ADT) *p* = 0.004 10 yr OS 74% (ADT + EBRT) vs. 55% (ADT) *p* < 0.001
Kaplan et al. (2013) [30]	Retrospective, SEER	577	EBRT vs. no EBRT (both groups received ADT evenly)	1995–2007	No benefit of adjuvant EBRT: OM 5.35 (EBRT) vs. 3.77 (no EBRT) events per 100 person-years, *p* = 0.193 PCSM 2.39 (EBRT) vs. 1.30 (no EBRT), *p* = 0.354
Abdollah et al. (2014) [31]	Retrospective, two institutions	1107	ADT vs. ADT + EBRT	1988–2010	Favours ADT with adjuvant EBRT 8 yr OM-free survival of 88% (ADT + EBRT) vs. 75% (ADT) (*p* < 0.01) 8-yr CSM-free survival 86% (ADT + EBRT) vs. 92%(ADT) (*p* = 0.08)
Rusthoven et al. (2014) [15]	Observational, SEER database	2991 (pN1 after staging LND)	Local therapy (RP, EBRT or both) vs. no local therapy	1995–2006	Favours local therapy over no local therapy 10 yr OS rate 65% vs. 42% (*p* < 0.001) 10 yr PCSS 78% vs. 56% (*p* < 0.001)
Wong et al. (2016) [27]	Retrospective, NCDB	7225	No adjuvant therapy vs. ADT alone EBRT alone ADT + EBRT	2004–2011	Favours adjuvant EBRT + ADT 5-yr OS rate, 85.2% (no therapy), 82.9% (ADT), 88.3% (EBRT), 88.8% (ADT + EBRT) (*p* ≤ 0.001)
Jegadeesh et al. (2016) [32]	Retrospective, NCDB	826	ADT + EBRT ADT alone	2003–2011	Results favour ADT + EBRT Improved OS (HR 0.67, 95% CI 0.55–0.83, *p* < 0.001)
Van Hemelryk et al. (2016) [33]	Retrospective, case-matched	69	Case-matching of pN1 and pN0 after EBRT + ADT	2006–?	5-yr bRFS 65% (pN1) vs. 79% pN0 (*p* = 0.08) 5-yr cRFS 70% (pN1) vs. 83% (pN0) (*p* = 0.04) 5-yr PCSS 92% (pN1) vs. 93% (pN0) (*p* = 0.66) 5-yr OS 82% (pN1) vs. 80% (pN0) (*p* = 0.58)
Poelaert et al. (2016) [34]	Retrospective	154	ADT + whole pelvis EBRT	2000–2016	5-year CSS 96% (±2%) 5-yr bRFS 67% (±5%) 5-yr cRFS 71% (±5%) 5-yr OS 89% (±3%)
Touijer et al. (2018) [5]	Retrospective, three institutions	1338	Observation vs. ADT alone vs. ADT + EBRT	1988–2010	Favours adjuvant ADT + EBRT over ADT HR 0.46 for OS (95% CI 0.32–0.66, *p* < 0.0001) Favours adjuvant ADT + EBRT over observation HR 0.41 for OS (95%CI 0.27–0.64, *p* < 0.0001)
Gupta et al. (2019) [28]	Retrospective	8074	Observation ADT ADT + EBRT	2004–2013	Results favour adjuvant ADT+ EBRT over ADT HR 0.76 for OS (95%CI 0.63–0.93, *p* = 0.007) Results favour adjuvant ADT+ EBRT over observation HR 0.77 for OS (95%CI 0.64–0.94, *p* = 0.008)
EBRT as adjuvant treatment					
Fonteyne et al. (2022) [35]	RCT PROPER trial	69	Prostate-only radiotherapy (arm A) vs. whole-pelvis radiotherapy (arm B)	2016–2021	Favours WPRT 3 yr 88% (PORT) vs. 92% (WPRT) (*p* = 0.31) 3-yr bRFS 79% (PORT) vs. 92% (WPRT) (*p* = 0.08) 3-yr OS 92% (PORT) vs. 93% (WPRT) (*p* = 0.61)
Tilki et al. (2021) [36]	Retrospective, five institutions	1491	Adjuvant vs. early salvage radiation therapy	1989–2016	Favours adjuvant EBRT in case of pN1 (HR 0.66, 95%CI 0.44–0.99; *p* = 0.04)
Chemotherapy as adjuvant treatment					
Ahlgren et al. (2018) [37]	RCT SPCG-12 trial	55/459 (27 arm A and 28 arm B)	Arm A: docetaxel Arm B: surveillance	2005–2010	No difference in time to BCR > 0.5 ng/mL (*p* = 0.06)

Abbreviations: pN1M0 patients = patients with pathologically node-positive disease in postoperative setting, either after initial treatment with radical prostatectomy (RP) and lymph node dissection (LND) or after staging LND; ADT = androgen deprivation therapy; EBRT = external beam radiotherapy; LT = local therapy (i.e., EBRT or RP); SEER = Surveillance, Epidemiology, and End Results; NCDB = National Cancer Database; OS = overall survival; CI = confidence interval; HR = hazard ratio; BCR = biochemical recurrence; CSS = cancer-specific survival; OM = overall mortality; PCSM = prostate cancer-specific mortality; CSM = cancer-specific mortality; PCSS = prostate-cancer-specific survival; bRFS = biochemical relapse-free survival; cRFS = clinical relapse-free survival; ACM = all-cause mortality.

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
