# Peer review of "Treating Primary Node-Positive Prostate Cancer: A Scoping Review of Available Treatment Options"

_cancers, 2023, doi:10.3390/cancers15112962_

Round 1

Reviewer 1 Report

English language should be improved in bothgrammar and syntax

Lines 144-147 refer to ADT therapy in associationwith surgical therapy and radiotherapyreaching the conclusion that the association of one or the otherdoes not give statistically significant differencesHowever, the age of the patients undergoing the treatments is not reportedtherefore I suggest thatyou investigate this aspect further

Paragraph 3.2.4 at line 231 discusses the use of docetaxel chemotherapy as an adjuvantat thisregart i suggest the following articlehttps://pubmed.ncbi.nlm.nih.gov/32474391/

It is possible to analyze the comorbidities of patientswith prostate cancer to understand whether they act as protective factors or risk factors for the progression of oncological disease in patients whoundergo ADT. At this regard the following articlecould be interestinghttps://pubmed.ncbi.nlm.nih.gov/32570240/

Minor revisions

Reviewer 2 Report

Nice review for patients with cN1/pN1 prostate cancer; Gives the reader an overview about potential treatment options. However, I missed a conclussion and recommendation which treatment would be best for the patient. Moreover, due to the introduction of new drugs like apalutamide, duralutamide in the course of treatment and modern diagnostic procederures like PSMA PET scan, a change and update in treatment options must be assumed in the near future.

Round 2

Reviewer 1 Report

Authors answered all comments and suggestions.

Minor revisions.

Reviewer 2 Report

I am fine with the revised manuscript